# HBM4EU Diisocyanates Study—Research Protocol for a Collaborative European Human Biological Monitoring Study on Occupational Exposure

**DOI:** 10.3390/ijerph19148811

**Published:** 2022-07-20

**Authors:** Kate Jones, Karen S. Galea, Bernice Scholten, Marika Loikala, Simo P. Porras, Radia Bousoumah, Sophie Ndaw, Elizabeth Leese, Henriqueta Louro, Maria João Silva, Susana Viegas, Lode Godderis, Jelle Verdonck, Katrien Poels, Thomas Gӧen, Radu-Corneliu Duca, Tiina Santonen

**Affiliations:** 1Health & Safety Executive, Buxton SK17 9JN, UK; liz.leese@hse.gov.uk; 2Institute of Occupational Medicine (IOM), Edinburgh EH14 4AP, UK; karen.galea@iom-world.org; 3Organization for Applied Scientific Research (TNO), NL-3700 AJ Zeist, The Netherlands; bernice.scholten@tno.nl; 4Finnish Institute of Occupational Health (FIOH), Työterveyslaitos, FI-00032 Helsinki, Finland; marikaloikala@gmail.com (M.L.); simo.porras@ttl.fi (S.P.P.); tiina.santonen@ttl.fi (T.S.); 5French National Research and Safety Institute, 54500 Vandoeuvre-les-Nancy, France; radia.bousoumah@inrs.fr (R.B.); sophie.ndaw@inrs.fr (S.N.); 6National Institute of Health Dr. Ricardo Jorge (INSA), 1649-016 Lisbon, Portugal; henriqueta.louro@insa.min-saude.pt (H.L.); m.joao.silva@insa.min-saude.pt (M.J.S.); 7Centre for Toxicogenomics and Human Health, NOVA Medical School, Universidade NOVA de Lisboa, 1169-056 Lisbon, Portugal; 8Public Health Research Centre, NOVA National School of Public Health, Universidade NOVA de Lisboa, 1600-560 Lisbon, Portugal; susana.viegas@ensp.unl.pt; 9Comprehensive Health Research Center (CHRC), 1169-056 Lisbon, Portugal; 10Centre for Environment and Health, Department of Public Health and Primary Care, KU Leuven (University of Leuven), 3000 Leuven, Belgium; lode.godderis@kuleuven.be (L.G.); jelle.verdonck@kuleuven.be (J.V.); katrien.poels@kuleuven.be (K.P.); radu.duca@lns.etat.lu (R.-C.D.); 11IDEWE, External Service for Prevention and Protection at Work, 3001 Heverlee, Belgium; 12Institute and Outpatient Clinic of Occupational (IPASUM), Friedrich-Alexander-Universität Erlangen-Nürnberg, 91054 Erlangen, Germany; thomas.goeen@fau.de; 13Unit Environmental Hygiene and Human Biological Monitoring, Department of Health Protection, Laboratoire National de Santé (LNS), 3555 Dudelange, Luxembourg

**Keywords:** diisocyanates, regulation, biomarkers, exposure assessment, workers

## Abstract

Diisocyanates have long been a leading cause of occupational asthma in Europe, and recently, they have been subjected to a restriction under the REACH regulations. As part of the European Human Biomonitoring project (HBM4EU), we present a study protocol designed to assess occupational exposure to diisocyanates in five European countries. The objectives of the study are to assess exposure in a number of sectors that have not been widely reported on in the past (for example, the manufacturing of large vehicles, such as in aerospace; the construction sector, where there are potentially several sources of exposure (e.g., sprayed insulation, floor screeds); the use of MDI-based glues, and the manufacture of spray adhesives or coatings) to test the usability of different biomarkers in the assessment of exposure to diisocyanates and to provide background data for regulatory purposes. The study will collect urine samples (analysed for diisocyanate-derived diamines and acetyl–MDI–lysine), blood samples (analysed for diisocyanate-specific IgE and IgG antibodies, inflammatory markers, and diisocyanate-specific Hb adducts for MDI), and buccal cells (micronucleus analysis) and measure fractional exhaled nitric oxide. In addition, occupational hygiene measurements (air monitoring and skin wipe samples) and questionnaire data will be collected. The protocol is harmonised across the participating countries to enable pooling of data, leading to better and more robust insights and recommendations.

## 1. Introduction

HBM4EU (www.hbm4eu.eu, date accessed 13 July 2022) is a collaborative project designed to coordinate and advance human biomonitoring in Europe and to provide evidence for chemical policy making. Within the project there is a specific work strand focussed on occupational exposures. Under this work strand, research has already been conducted looking at hexavalent chromium exposures across three sectors (plating, welding, and surface coating) in nine countries (Belgium, Finland, France, Italy, Luxembourg, Poland, Portugal, The Netherlands, and the United Kingdom) [1,2]. That research successfully coordinated a harmonised approach to undertaking the sampling during site visits and guaranteed comparability of analysis through the use of the HBM4EU quality assurance scheme. The study showed the benefit of such an approach, allowing the pooling of data from different institutes and countries, thereby increasing the power of the study and the strength of any conclusions. Of course, there were points that could be improved; these were reflected on and recently reported [3] for the benefit of future initiatives.

This paper presents the protocol for another study that was conducted within the occupational exposure work strand of HBM4EU, looking at diisocyanate exposures. This study benefits from many of the same collaborators as the chromates study previously reported, who contributed to study planning, design and standard operating procedures (SOPs), and seeks to address some of the learning points from that experience. A further occupational study on e-waste exposures is also planned, and the protocol has been published [4].

Diisocyanates have long been a leading cause of occupational asthma [5,6]. Exposure limits are generally very low (in the µg/m^3^ range, https://limitvalue.ifa.dguv.de/, date accessed 13 July 2022), and the European Commission’s Advisory Committee on Safety and Health at Work (ACSH) recently proposed a binding occupational exposure limit of 10 µg/m^3^ (measured as –NCO, total reactive isocyanate groups), reducing it to 6 µg/m^3^ from 1 January 2029 (https://circabc.europa.eu/ui/group/cb9293be-4563-4f19-89cf-4c4588bd6541/library/0d11d394-b1e8-4e1a-a962-5ad60f4ab2ae/details, date accessed 13 July 2022). Engineering controls and personal protective equipment (PPE) are often required to enable compliance with these low limits. In many countries, health surveillance is also mandated due to the known health impacts (https://www.finlex.fi/en/laki/kaannokset/2001/en20011485.pdf, date accessed 13 July 2022) [7], and this is also recommended in the ACSH’s opinion. Diisocyanates are used in many industrial applications including coatings, adhesives and lacquers, in moulding, and in slab foam production [6]. They are highly reactive chemicals with fast cure times, making them attractive to industry and hard to replace; these features also mean that any substitute chemicals with suitable properties will potentially demonstrate similar health impacts. In light of the health concerns and the difficulty, currently, of substitution, the European Commission has adopted a restriction within the Registration, Evaluation, Authorisation and Restriction of Chemicals (REACH) regulation to limit the use of diisocyanates in industrial and professional applications (Commission Regulation (EU) 2020/1149 https://eur-lex.europa.eu/legal-content/EN/ALL/?uri=uriserv:OJ.L_.2020.252.01.0024.01.ENG, date accessed 13 July 2022). This restriction (adopted in August 2020) requires technical and organisational control measures to be implemented and mandates a minimum standardised training course for workers (https://www.safeusediisocyanates.eu/, date accessed 13 July 2022). There is currently no proposed Biological Limit Value for diisocyanates within the EU [8], due to the absence of an exposure limit and uncertainty about the correlation between inhalation exposure and urinary metabolite levels. This study seeks to provide additional information in this area. Individual countries within Europe do have guidance values for diisocyanates [9,10], and these will be valuable in interpreting the results from this study.

In order to inform the current proposed study, a systematic review on the published biomonitoring data on exposure to diisocyanates was undertaken [11]. The review found that about half of the studies were published were prior to 2010 and, hence, might not reflect current workplace exposure. There was also a large variability within and between studies and across sectors, which could be potentially explained by several factors including worker or workplace variability, short half-lives of biomarkers, and differences in sampling strategies and analytical techniques. Several studies addressed exposure to toluene diisocyanate (TDI) in flexible foam production, methylene diphenyl diisocyanate (MDI) in polyurethane production, or hexamethylene diisocyanate (HDI) in paint spraying of vehicles in spray booths. However, fewer data were available on the occupational exposure to diisocyanates in the manufacturing of other vehicles such as aerospace, shipping and large commercial vehicles, the construction sector, where there are potentially several sources of exposure (e.g., sprayed insulation, floor screeds), the use of MDI-based glues, and the manufacture of spray adhesives or coatings [11]. A recent review [6] also identified these sectors as potentially significant, based on airborne measurements of diisocyanate exposure (although several of these also have dermal contact potential). The current HBM4EU diisocyanates study, therefore, aimed to include these sectors, to harmonise the sampling strategy and ensure comparable analytical techniques.

Urinary diamines are the most commonly used biomarkers for exposure to diisocyanates [11]. However, these are not specific to diisocyanates; the corresponding diamines (hexanediamine (HDA), toluenediamine (TDA), and methylene dianiline (MDA)) are also commonly used industrial chemicals. The study will, therefore, also investigate other biomarkers such as urinary lysine adducts, diisocyanate-specific haemoglobin adducts, and diisocyanate-specific IgG, as well as some potential effect biomarkers, including diisocyanate-specific IgE and IgE antibodies, a panel of inflammatory markers, fractional exhaled nitric oxide (FeNO), and micronuclei (MN) in buccal cells.

The main objectives of the study are:To provide new data on the exposure to diisocyanates in specific sectors based on a harmonized sampling protocol applied across five European countries;To test the usability of different biomarkers in the assessment of exposure to diisocyanates;To provide background data to support the implementation of an EU-wide limit value for diisocyanates and to allow for the follow-up of the effectiveness of the diisocyanates REACH restriction.

In this manuscript, we provide an outline of the research protocol for the study.

## 2. Materials and Methods

The various sampling aspects of the study are described by SOPs, listed in Table 1 and available as Appendix A, which are also available from the HBM4EU website (https://www.hbm4eu.eu/online-library/, date accessed 13 July 2022) (under “Guidelines, protocols and questionnaires” then “Materials for the occupational studies under HBM4EU”). As described previously, these were developed from those used in the HBM4EU chromates study, whilst also being revised to reflect the experiences from that study.

### 2.1. Company and Workers Recruitment

The aim was to focus on recruitment of small- and medium-sized enterprises (SMEs) using diisocyanates in specific industry sectors. The industrial sectors of interest included manufacturing and repair of large vehicles (non-booth spraying of, e.g., boats/planes), the use of diisocyanate-based hot-melt glues in different sectors, and the construction sector, which includes different sources of diisocyanates exposure (floorings/screeds, insulation).

For this study, results in the exposed worker group will be compared to non-exposed workers (e.g., administration staff) within the recruited companies (control subjects). Each participating country (Belgium, Finland, France, The Netherlands, and United Kingdom) aimed to collect samples from 50 exposed and 25 control subjects, thus yielding a study population of 250–300 exposed workers and up to 150 controls. These numbers are in line with target numbers for other surveys within this project [1,4].

Each participating country identified and contacted suitable companies (that could fulfil the target population) from a variety of means including previous collaboration, direct approach, and through trade associations. Recruitment of the companies and workers followed the SOP developed under HBM4EU for the selection of participants, recruitment, informing participants, and obtaining informed consent^3^ (see Appendix A). Interested companies received a company information leaflet explaining the aims and objectives of the study and what would be expected from them and their workers through their participation. Workers involved in the activities of interest were invited by the researchers to participate. Workers’ participation was entirely on a voluntary basis. Hence, all communication with workers was arranged in order to secure as free a choice of participation for the worker as possible. A participant information leaflet for the workers was distributed and discussed during the first contact with the workers. Workers completed a certificate of informed consent if they decided to participate. The same approach was followed for control subjects.

Common information leaflets and informed consent forms, developed under HBM4EU, were translated and provided in the national languages. This approach ensures that all the participating companies and workers receive the same relevant information on the study. National information, including information on specific national legislation and the contact details of the relevant national research group, was also added.

Study protocols and information materials have been submitted for approval by medical/research ethics boards in each of the participating countries (Table 2). The informed consent forms will be archived for the entire study duration and not less than 5 years.

### 2.2. Company and Worker Questionnaires

As for the HBM4EU chromates study [1], two questionnaires will be used to collect relevant contextual information from the study participants (see Appendix A). One is a self-administered questionnaire to be completed by a company representative, prior to the sampling campaign. The second is to be completed by the researcher while interviewing the worker as close as possible to the end of work shift.

The company questionnaire collects general background information on the company including some details regarding general training, exposure monitoring, and occupational health and safety practices. Details of the operational conditions related to their use of diisocyanates within a targeted sector (as applicable) are obtained through questions on, for example, the identity and amount of diisocyanate used, frequency of operations, size of the worked parts, and number of involved workers.

The interviewer-led post-shift worker questionnaire is more detailed. Different questionnaires have been prepared for workers involved in the use of adhesives, coating large surfaces, spraying foam, spraying large vehicles, and flame-cutting, welding, or grinding diisocyanate-coated items. Possible background exposures from sources other than workplaces are investigated through the questions related to the living environment, personal habits (e.g., smoking, diet), as well as recreational activities or hobbies that may lead to diisocyanate exposure (e.g., home renovation or car spraying). Questions on respiratory health are also covered through an adaptation of HSE’s respiratory questionnaire from the UK, itself a hybrid of the original Medical Research Council (MRC) questionnaire [12], and that used by the European Community Respiratory Health survey (ECRHS) [13]. Due to the proposed timing of the fieldwork, additional questions have also been added regarding COVID-19; these are based on the OMEGA-NET/EPHOR questionnaire.

Job descriptions are addressed through questions concerning the characteristics of the specific tasks. A detailed description of the tasks performed on the day of providing biological samples is collected through questionnaire sections specific to each sector, including their duration in a work shift and frequency, details of the risk management measures (RMMs) used during the work activities, e.g., local exhaust ventilation (LEV), PPE, information and training, and personal hygiene practices, as well as the occurrence of incidents during the shift.

### 2.3. Blood Sampling and Analysis

Blood sampling requires a clean and private space, trained phlebotomists, and appropriate sterile material for the collection [14]. One blood sample will be collected from each participant, both workers and control subjects, at any time during the workweek. The blood sample is distributed into two tubes: a 6 mL sodium heparin tube (Tube 1) and a 9 mL K_2_EDTA tube (Tube 2). Samples are stored at +4 °C during the onsite collection and transport to the laboratory. Tube 1 is centrifuged for 5 min at 2000× *g*. The plasma (supernatant) is transferred to a microtube and a cryotube (approximately 1.5 mL in each tube) and stored at −20 °C (microtube) and at −80 °C (cryotube) until analysis. Tube 2 is processed to isolate the red blood cells (as described by [15]) and stored at −20 °C until analysis.

Tube 1 samples are analysed for diisocyanate-specific IgE and IgG antibodies (microtube) and for the inflammatory markers IL-6, IL-8, TNF-alpha, IFN-gamma, TIMP-1, and MMP-9 (cryotube). Tube 2 samples are analysed for diisocyanate-specific Hb adducts for MDI. Each biomarker analysis for the blood samples will be undertaken by a single laboratory to standardise bias and minimise variability, as there are no quality assurance schemes for these assays available within HBM4EU. To determine inflammatory markers, plasma samples will be analysed with the MSD inflammatory cytokine panel (Meso Scale Diagnostics, Rockville, MD, USA), according to the manufacturers’ protocol [16]. HDI and MDI specific IgE and IgG contents in plasma are determined with immunoCAP fluoroenzyme immunoassay methods according to instructions given by the manufacturer (Thermo Fisher Scientific Inc., Vantaa, Finland). Haemoglobin adducts of 4,4′-methylene diphenyl diisocyanate in human blood will be analysed according to the method of Gries and Leng [17]. Globin is isolated and then hydrolysed using hydrochloric acid and heat. Samples are then cooled, made alkaline, and extracted into trichloromethane. The extract is evaporated, derivatised with heptafluorobutyric anhydride, then re-evaporated before reconstitution in ethyl acetate. Analysis is by gas chromatography–high resolution mass spectrometry.

### 2.4. Urine Sampling and Analysis

Urine samples for workers are collected three times: before the work shift (pre-shift void), at the end of the work shift (post-shift void), and the next day in the morning (first morning void). For the control subjects, only one sample is collected towards the end of the shift.

Workers are asked to remove their work clothes (overalls) and to wash their hands thoroughly with soap and water before providing urine samples to avoid contamination of samples. Printed instructions on the procedure and hygiene of urine sample collection are given to each participant.

The urinary diamines (TDA, HDA, and MDA) will be analysed by institutes that have successfully participated in the HBM4EU ICI/EQUAS scheme (that took place between December 2018 and July 2020) using gas chromatography or liquid chromatography and mass spectrometry [18,19].

In addition, samples from MDI-exposed workers (and controls) will be analysed for acetyl–MDI–lysine as a potentially more specific urinary biomarker. Diisocyanate–lysine adducts have been reported in plasma samples of workers [20] but, to date, there are no reports of measuring this metabolite in urine. A single laboratory will undertake the analysis to standardise bias and minimise variability, as there are no quality assurance schemes for this assay available within HBM4EU.

### 2.5. Fractional Exhaled Nitric Oxide (FeNO) Samples

Nitric oxide is produced by the lungs and is present in the exhaled breath of all humans. FeNO is the fractional concentration of exhaled nitric oxide in the gas phase of exhaled air. The collection of FeNO samples is a non-invasive technique that is simple to perform and causes no ill health effects to people with existing respiratory conditions [21]. The American Thoracic Society recommends the use of FeNO to support the diagnosis of asthma and continued monitoring of respiratory inflammation and its response to anti-inflammatory therapy.

The collection and analysis of FeNO samples in this project is intended to further the understanding of diisocyanate exposures and their effects on workers and to explore FeNO’s potential as a health effect biomarker.

To enable standardisation amongst the different teams collecting FeNO in this project, the NIOX VERO (www.niox.com, date accessed 13 July 2022) has been chosen for the collection and measurement of samples. The NIOX VERO is a small, handheld, battery powered device. It requires a ten-second exhalation of breath by an individual. The last 3 s of the 10 s exhalation are analysed by the electrochemical sensor to provide a definitive FeNO result in parts per billion (ppb) [22]. The instrument has in-built calibration, and positive and negative controls are run before starting sampling. The choice of device was by convenience and does not imply any endorsement by the study team.

FeNO samples will be collected from workers in line with the urine sampling, i.e., three times (1) before the work shift (pre-shift), (2) at the end of the work shift (post-shift), and (3) the next day in the morning. For control subjects, one sample is collected towards the end of the work shift (when a urine sample is also collected). For each sample, information is recorded on whether the worker has eaten, drunk, or smoked a cigarette up to an hour prior to the sample collection.

### 2.6. Buccal Cells Sampling

The buccal micronucleus (MN) assay is a minimally invasive approach for measuring DNA damage, cell proliferation, cell differentiation, and cell death in exfoliated buccal cells [23]. It offers a great opportunity to evaluate, in a clear and precise way, the appearance of genetic damage, whether it is present as a consequence of occupational or environmental risk. It is also reliable, fast, relatively simple, cheap, and minimally invasive and causes no pain [24].

Previous studies on workers exposed to diisocyanates used in the manufacture of polyurethane foam have reported increased genotoxic effects [25]. Both TDI- and MDI- exposed workers showed increased frequencies of MN in peripheral blood lymphocytes [26,27] and in buccal epithelial cells [27], the latter confirming the suitability of this assay to assess early effects caused by diisocyanates.

Buccal cell samples are collected by rotating a small-headed toothbrush 10 times against the inside of both cheek walls in a circular motion, starting from the middle and gradually increasing in circumference to produce an outward spiral effect. The head of the brush is then placed into the fixative container (Saccomanno’s fixative—50% alcohol which contains approximately 2% Carbowax 1540 (Merck Life Science UK Limited, Gillingham, UK)) and rotated such that the cells are dislodged and released into the suspension [23]. Samples are then transported to the laboratory and stored at +4 °C until analysis. A single laboratory will undertake the analysis to standardise bias and minimise variability, as there are no quality assurance schemes for this assay available within HBM4EU.

### 2.7. Occupational Hygiene Measurements

In order to characterise the external exposure, a number of occupational hygiene measurements are collected. A proportion of workers (where inhalation exposures are considered to be potentially significant) will have personal air monitoring conducted using the Supelco ASSET™ EZ4 Dry Samplers (Merck Life Science UK Limited, Gillingham, UK)) for Isocyanates (https://theanalyticalscientist.com/fileadmin/tas/issues/App_Notes/Sampling_Analysis_Isocyanates.pdf, date accessed 13 July 2022). Although the sampler does not facilitate a strict “total reactive isocyanate group” (–NCO) measurement, as required by some countries’ occupational exposure limit, e.g., the UK [10], it does make standardisation of sampling, analysis, and calibration standards much easier to achieve. It is proposed that air sampling will only take place during activities where aerosols are likely, such as spraying and welding/grinding activities. Sampling will be conducted at 250 mL/min for up to four hours, except for short-duration tasks (15 min) where a sampling rate of 850 mL/min is recommended (to capture sufficient sample). Characterisation of bulk samples to estimate the total –NCO present will be recommended where it is likely that the available calibration standards do not reflect >80% of the –NCO within the product. Further information on whether respiratory protective equipment (RPE) was used, at what time the exposure took place, and what the duration was (e.g., 11:00 a.m., 240 min) is recorded.

In addition to air monitoring, some hand wipe samples will also be collected using the Swype^TM^ colorimetric hand wipes (SKC Ltd., Blandford Forum, UK) for either aliphatic or aromatic diisocyanates, as appropriate for the sampled activity [28]. Samples will be taken pre- and post-task for relevant activities such as use of coatings and spray applications. The palmar region of the dominant hand will be wiped three times in a circular fashion. The wipe is then placed in the developer solution and, after four minutes, any colour change is noted and the wipe is photographed next to an interpretative scale. The extent of dermal contamination will be judged manually on a four-point scale, and each research institute will be provided with a graded colour chart. It is planned to undertake a round-robin exercise of sharing photographs and grading them to ensure some consistency in scoring across the participating institutes.

### 2.8. Exposure Modelling

Many different approaches can be used for estimating occupational exposure to chemical substances, with the use of predictive exposure models becoming more frequent. When applying exposure assessment modelling tools, users are required to select options from several possible input parameters. Hence, results obtained with the tools could be affected by factors such as the professional experience and judgment of the tool user and access to an appropriate level of information.

To explore this further, a standardised proforma will be used to collect contextual information, e.g., details of the RMMs in place and being used, operational conditions, materials generated and used, etc., about the work activities observed; from these exposures, scenarios will be generated. Participants with differing knowledge about the workplace environments and activities will be given the generated workplace exposure scenarios and asked to use a selected REACH model, ART 1.5, via the TREXMO (TRanslation of EXposure MOdels) tool to estimate inhalation exposure (https://trexmo.unisante.ch/, date accessed 13 July 2022).

Comparisons of the exposure estimates generated between the different types of users will be made, with these estimates also being compared with the actual exposure measurement results.

### 2.9. Data Handling and Communication

Data generated by the different research institutes will be entered into an Excel template where reporting is constrained to ensure that data are recorded in a consistent manner. Information on such a limit of detection for each assay is also recorded. All datasets will then be combined and analysed centrally according to a developed data analysis plan (https://www.hbm4eu.eu/work-packages/deliverable-10-7-revised-data-management-plan-dmp/, date accessed 13 July 2022). Individual research institutes are responsible for reporting back results to individual workers and to participating companies.

The communication plan is designed to ensure that all the relevant results are communicated to all study participants (companies, workers, and, equally, to controls) and national and EU stakeholders. Dissemination of information to the scientific community is also covered. The results will be presented in a different manner dependent on the type of participant (e.g., individual levels for workers or aggregated levels to companies). Effective and consistent communication with participants and stakeholders will help ensure that the results obtained are accessible to support organizational learning and used for decision-making. This will occur at the companies’ and workers’ levels, but also at the regulatory level. Communication is key to guaranteeing that the tools and data being developed in the scope of HBM4EU will be available to be used by risk assessors.

## 3. Conclusions

This manuscript describes the methodology followed in a harmonised multicentre study of occupational exposure to diisocyanates. Five European countries are involved, collecting biomonitoring and occupational hygiene data in understudied activities. Besides the generation of useful data for the EU and other regulatory decision-making bodies, this study will allow exploration of more sensitive and/or specific biomarkers for the biomonitoring of exposure to diisocyanates. The sampling campaign is now completed and the reporting of the results is expected by the end of 2022.

## Figures and Tables

**Table 1 ijerph-19-08811-t001:** Standard Operating Procedures (SOPs) developed for the HBM4EU diisocyanates study.

SOP No.	Title
1	Selection of participants, recruitment, information to the participants, informed consent
2	Completion of company and worker questionnaires
3	Blood sampling, including sample storage and transfer
4	Urine sampling, including sample storage and transfer
5	Assessing dermal exposure to diisocyanates
6	Air sampling of diisocyanates
7	Breath sampling for fractional exhaled nitric oxide (FeNO)
8	Buccal cells sampling, including sample storage and transfer
9	Comparing occupational hygiene measurements with estimates generated using exposure models
10	Communication plan for results

**Table 2 ijerph-19-08811-t002:** Details of ethical approval in each country.

Country	Reference	Date Approved	Ethics Committee
Belgium	S-64670	28 September 2020 (for study commencement)	Ethische Commissie Onderzoek UZ/KU Leuven, Belgium
Finland	HUS/2237/2020	12 August 2020	Coordinating ethics committee, HUS Joint Authority, Helsinki, Finland
France	2020-A01308-31	8 September 2020	Comité de Protection des Personnes (CPP) “Ile de France II”
The Netherlands	2021-007	22 February 2021	Medical ethical review board CMO Regio Arnhem-Nijmegen
United Kingdom	20/EM/0188	15 September 2020	NHS East Midlands-Derby Research Ethics Committee

## Data Availability

No new data were created or analysed in this protocol. Data sharing is not applicable to this article.

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
