# Peer review of "HBM4EU Diisocyanates Study—Research Protocol for a Collaborative European Human Biological Monitoring Study on Occupational Exposure"

_ijerph, 2022, doi:10.3390/ijerph19148811_

Round 1
Reviewer 1 Report
The manuscript entitled „HBM4EU diisocyanates study – research protocol for a collaborative European human biological monitoring study on occupational monitoring” is a very interesting article. Diisocyanates are subject of the REACH Regulation and according to CLP Regulation the have harmonised classification which was omitted in the manuscripted. The predominant health effects of the diisocyanates of occupational exposure are sensitisation and irritation of skin of respiratory tract caused by acute and long-term exposure.
Special comments:
11. In the Introduction section I would advice to add information concerning in which countries the hexavalent chromium study was conducted.
22. In this section, additionally, would be good looking information that Committee for Risk Assessment do not reccommend to set BLV, but you have made the attempts. Moreover, In the Supplementary Material, I have not seen the ATSDR materials. Have the Authors use it during preparation the manuscript?
33. In the Standard Operating Procedures there is „Assessing dermal exposure to diisocynates”. How the Authors do this? By which computer programme?
44. In the company and workers recruitment there are missing data how the procedure was conducted. It should be clearly stated and written how many companies took part in the study.
55. Idea for the next article. Maybe it should be worth to consider to compare the empirical data from the ART programme from the workplace exposure scenarios with the analytical data resulted from the tests.
16. Does the study have the limitations? In my opinion the Authors do not have to compare the resulted data with the BLV because in EU law or any other legal act because it has not elaborated yet.
Author Response
Please see attached response.

Reviewer 2 Report
This research protocol is of great interest to the scientific community, business and workers. Occupational exposure to certain agents, in this case diisocyanates, has a very negative impact on workers' health. It is therefore imperative that maximum exposure limits are established in order to guarantee the health of workers.
The introduction is well written and frames the study clearly.
The study design seemed to me to be quite detailed and appropriate.
I'm looking forward to the results.
Author Response
We thank the reviewer for taking the time to review our manuscript and their very positive comments.
Reviewer 3 Report
The reviewed paper is a description of the research methodology contained in the HBM4EU epidemiological research project, which received funding and, as the content of the work shows, has already been partially implemented. Therefore, it is not justified to assess the correctness of the adopted research methodology and its scope.
In the comments to my review, I indicated what should be improved.
The reviewed paper describes the procedures that will be performed under the project that has received funding. Such tests are important in occupational exposure assessment and should be performed more frequently.
Strengths:
- clear formula of the presented content
- correct protocols for data collection
- correctly designed research (questionnaires, number of participants, research schedule, types of the samples).
Weaknesses:
- text formatting - improve the way you write links to web pages, note IJERPH guidelines for the Authors
- not all web pages can be opened
- the number of countries where the research will be conducted is incorrect - 7 or 6?
- References - correct the formatting and add up-to-date articles related to the topic of the thesis.
- line 229: gas chromatography or liquid chromatography and mass spectrometry (this is the order how these coupled analytical techniques should be written).
- Table 2 - provide the names of the Ethics Committee in English or in the language of the country where the research is to be performed.
Author Response
Please see attached response.

Reviewer 4 Report
Review of HBM4EU Isocyanates-Study Research Protocol by Kate Jones et al.
Industrial isocyanates are used in many industries are obviously a major trigger for the development of asthma and COPD. A strength of this study is that it employs many types of sampling blood, urine, air, and breath sampling for isocyanates, along with company and worker questionnaires. The proposed sample size of 300 exposed workers and 150 controls will hopefully be large enough to yields statistically robust results. The methodology of this study appears to be sound, although I would also like to read what the other reviewers say. I think this paper will be a useful addition to the literature. I have a few suggestions which may be helpful.
SELECTION OF NATIONS- Comprised of Belgium, Finland, France, Netherlands, United Kingdom. All in Northern Europe. Would data apt to be similar in Southern European Nations such as Spain, Portugal, Italy, and Greece or the large industrialized nation of Germany?
NUMBER OF NATIONS On line 27 you state that 6 nations are involved- is this correct?
SELECTION OF COMPANIES- Will the types of companies selected be reported in the study itself.
QUESTIONAIRE –OTHER EXPOSURES- Will the questionare ask about other exposures related to asthma/ COPD like first/ second hand smoke, outdoor air pollutants, pesticides, solvents, indoor mold water damage, and other industrial exposures
Author Response
Please see attached response.
